# The molecular mechanism of nuclear transport revealed by atomic-scale measurements

Loren E Hough[2†], Kaushik Dutta[3†], Samuel Sparks[1], Deniz B Temel[1], Alia Kamal[2], Jaclyn Tetenbaum-Novatt[2], Michael P Rout[2]*, David Cowburn[1]*

[1]Department of Biochemistry, Albert Einstein College of Medicine, Bronx, United States; [2]The Rockefeller University, New York, United States; [3]New York Structural Biology Center, New York, United States

**Abstract** Nuclear pore complexes (NPCs) form a selective filter that allows the rapid passage of transport factors (TFs) and their cargoes across the nuclear envelope, while blocking the passage of other macromolecules. Intrinsically disordered proteins (IDPs) containing phenylalanyl-glycyl (FG)-rich repeats line the pore and interact with TFs. However, the reason that transport can be both fast and specific remains undetermined, through lack of atomic-scale information on the behavior of FGs and their interaction with TFs. We used nuclear magnetic resonance spectroscopy to address these issues. We show that FG repeats are highly dynamic IDPs, stabilized by the cellular environment. Fast transport of TFs is supported because the rapid motion of FG motifs allows them to exchange on and off TFs extremely quickly through transient interactions. Because TFs uniquely carry multiple pockets for FG repeats, only they can form the many frequent interactions needed for specific passage between FG repeats to cross the NPC.

*For correspondence: cowburn@cowburnlab.org (DC); rout@rockefeller.edu (MPR)

†These authors contributed equally to this work

Competing interests: The authors declare that no competing interests exist.

## Introduction

Nuclear pore complexes (NPCs) are the sole mediators of bi-directional nucleocytoplasmic trafficking. Transport is rapid and reversible, with the entire process of transport factor (TF) docking, passage and release across the NPC taking only a few milliseconds (*Strawn et al., 2004*; *Hulsmann et al., 2012*). The NPC consists of a ~30-nm diameter central channel filled with phenylalanyl-glycyl-repeat-rich nucleoporins (FG Nups) which provide the selective filter. Depletion or deletion of the FG Nups results in leaky, non-selective barriers (*Strawn et al., 2004*; *Di Nunzio et al., 2012*; *Hulsmann et al., 2012*; *Funasaka et al., 2013*). Moreover, the FG domains in isolation facilitate selective passage of TFs through nanopores (*Jovanovic-Talisman et al., 2009*; *Kowalczyk et al., 2011*) or accumulation in hydrogels (*Frey et al., 2006*; *Frey and Görlich, 2007*; *Schmidt and Gorlich, 2015*).

It is generally accepted that TF interaction with FG repeats reduce the diffusional barrier to enable selective transport (*Vasu and Forbes, 2001*; *Rout et al., 2003*; *Suntharalingam and Wente, 2003*; *Zeitler and Weis, 2004*), though the molecular mechanism of TF passage through the NPC remains largely unknown. TFs may alter the properties of the FG permeability barrier. Mesoscale observations of TF-FG repeat interactions in vitro have shown that TFs can change the height of FG brushes on planar surfaces (*Lim et al., 2007*; *Eisele et al., 2010*; *Kapinos et al., 2014*; *Wagner et al., 2015*), modulate the transport of inert cargo (*Lowe et al., 2015*), assemble with FG Nups into large assemblies (*Lowe et al., 2015*), and inhibit amyloid hydrogel formation observed for some FG Nups (*Milles et al., 2013*). Whether these behaviors arise from changes in FG structure or as a result of the multivalent FG-TF interaction remains undetermined, as do their contributions to in vivo nuclear transport.

**eLife digest** Eukaryotic cells have a nucleus that contains most of the organism's genetic material. Two layers of membrane form an envelope around the nucleus and protect its contents from the rest of the cell's interior. However, this protective barrier must also allow certain proteins and nucleic acids(collectively called 'cargo') to move in and out of the nucleus.

Cargo molecules can pass through channel-like structures called nuclear pore complexes, which are embedded in the nuclear envelope. However, transport across this barrier is highly selective. While small molecules can pass freely through nuclear pore complexes, larger cargo can only be transported when they are bound to so-called transport factors. The nuclear pore complex is a large structure made up of more than 30 different proteins called nucleoporins. Like all proteins, nucleoporins are built from amino acids. Many nucleoporins contain repeating units of two amino acids, namely phenylalanine (which is often referred to as 'F') and glycine (or 'G'). These 'FG nucleoporins' are found on the inside of the nuclear pore complex and interact with transport factors to allow them to transit across the nuclear envelope.

Several models have been put forward to explain how FG nucleoporins block the passage of most molecules. But it was unclear from these models how these nucleoporins could do this while simultaneously allowing the selective and fast transport of nuclear transport receptors. There was also a major lack of experimental data that probed the behavior of FG nucleoporins in detail.

Hough, Dutta et al. have now used a technique called nuclear magnetic resonance spectroscopy (or NMR for short) to address this issue. NMR can be used to analyze the structure of proteins and how they interact with other molecules. This analysis revealed that FG nucleoporins never adopt an ordered three-dimensional shape, even briefly; instead they remain unfolded or disordered, moving constantly. Nevertheless, and unlike many other unfolded proteins, FG nucleoporins do not aggregate into clumps. This is because they are constantly changing and continuously interacting with other molecules present inside the cell, which prevents them from aggregating.

Hough, Dutta et al. also observed that the repeating units in the FG nucleoporins engaged briefly with a large number of sites or pockets present on the transport factors. These FG repeats can bind and then release the transport factors at unusually high speeds, which enables the transport factors to move quickly through the nuclear pore complex. This transit is specific because only transport factors have a high capacity for interacting with the FG repeats. These findings provide an explanation for how the nuclear pore complex achieves fast and selective transport. Further work is needed to see whether certain FG nucleoporins specifically interact with a particular type of transport factor, to provide preferred transport routes through the nuclear pore complex.

Several mesoscale models have attempted to explain how FG Nups prevent the passage of most macromolecules while allowing selective transport of TFs alone and with their cargo. The FG Nups have been proposed to form a selective barrier due to their reversible inter-chain cohesion (hydrogel [*Frey et al., 2006*; *Frey and Görlich, 2007*; *Ader et al., 2010*] and bundle models [*Gamini et al., 2014*]), entropic exclusion (virtual gating model) (*Rout et al., 2000, 2003*; *Lim et al., 2006*), collapse upon TF binding (reduction in dimensionality model) (*Peters, 2005*), or a combination thereof (forest model) (*Yamada et al., 2010*). These models differ in their predictions of FG behavior on their own (ranging from highly mobile to fully self-associated) and upon binding of TFs. For example, the hydrogel, forest and reduction in dimensionality models invoke large changes in FG Nup behavior upon TF interaction (*Lim et al., 2007*; *Eisele et al., 2010*; *Kapinos et al., 2014*; *Wagner et al., 2015*). Crucially, none of these models or current observations fully explains how transport can be both selective and rapid, as is seen in vivo.

These conflicting models of the mechanism of NPC selectivity remain (*Schmidt and Gorlich, 2015*) because of a lack of atomic-scale experimental data describing FG Nup behaviors and interactions. Therefore, we used nuclear magnetic resonance (NMR) techniques to provide a rich readout of FG Nup behaviors at atomic detail (*Figure 1A*). We defined a minimal system including FG attachment, FG repeat type, and TF interaction, the essential features necessary to recapitulate selective transport in vitro (*Jovanovic-Talisman et al., 2009*; *Kowalczyk et al., 2011*). We measured the physical state of FG repeats with and without TFs bound. Because environment strongly affects intrinsically disordered

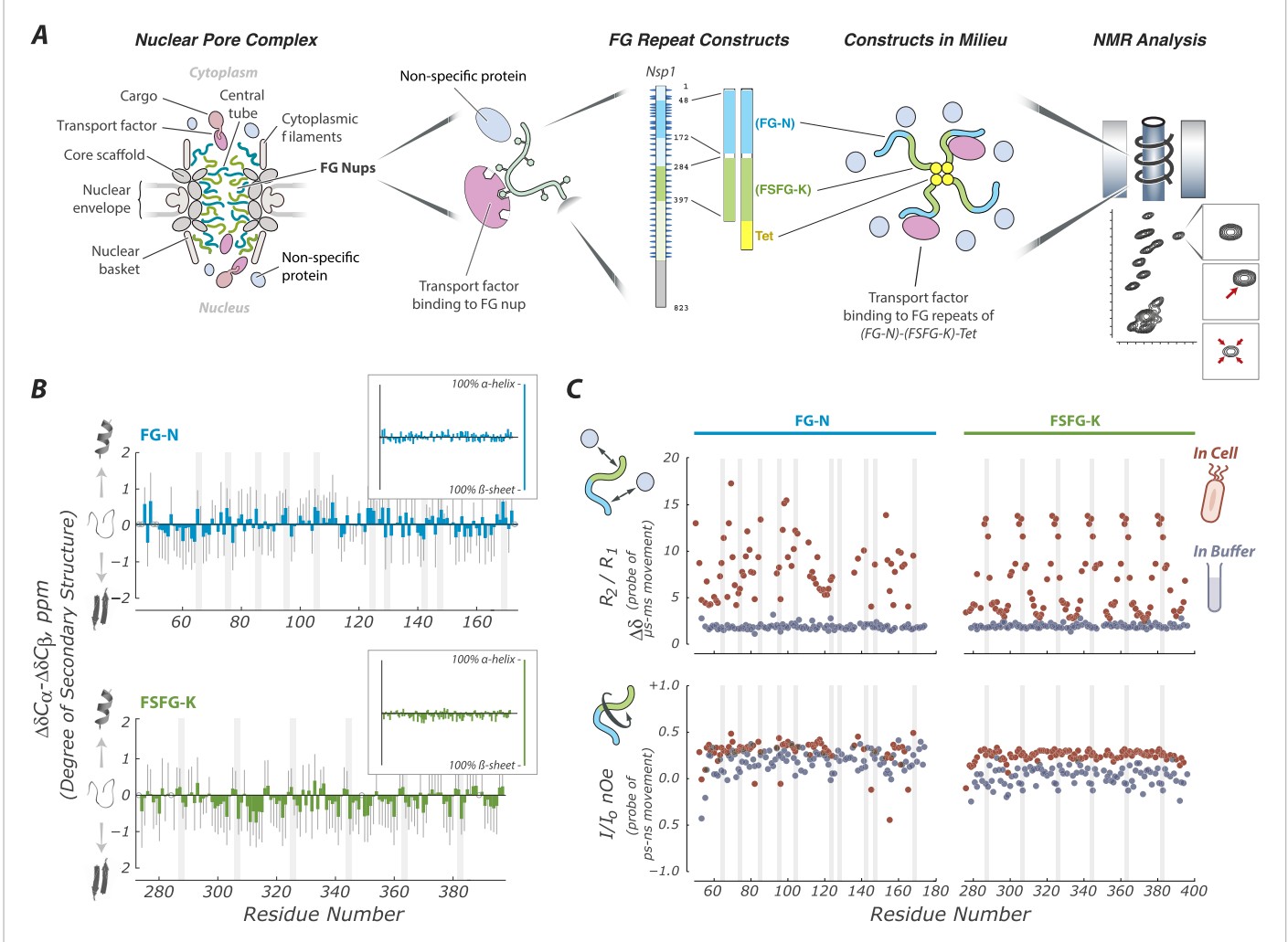

**Figure 1**. FG Nups are normally in a fully disordered and highly dynamic fluid state. (**A**) Our experimental approach includes key features of the NPC; a mixture of FG flavors, attachment at one end, and both specific (TF) and non-specific interactions with the cellular milieu. For example, our largest construct (FG-N-FSFG-K-tet) contains two fragments from Nsp1 (FG-N, turquoise; FSFG-K, green; full-length Nsp1 also shown with residue numbering), a separator (white) and the tetramerization domain of p53 (yellow). NMR analysis is performed on this construct and its variants, in milieu of various types; changes in position or intensity of peaks (bottom right) indicate changes in structure or interactions of the FG motifs. (**B**) Deviations of chemical shift values in cell (*Escherichia coli*) from predicted (colored bars) showing that FG-N and FSFG-K fragments are fully disordered, with no propensity for secondary structure. Also shown are standard errors of the mean (gray bars) and positions of FG motifs in the sequence (gray columns). Chemical shift values expected for an α-helix or β-sheet are approximately −4 and +4 ppm, respectively, as shown in the small insets. (**C**) The FG-N and FSFG-K constructs show significant interactions with the cellular milieu and exhibit very rapid motions. The upper panels show the ratio of $R_2/R_1$ indicative of overall motion and effects of multiple environments (chemical exchange) in cell (*E. coli*), compared to buffer A. The heteronuclear nuclear overhauser effect (nOe) is shown in lower panels, indicative of backbone motions. Gray columns indicate the locations of the FG motifs. Full experimental details and interpretations are available in 'Materials and methods' and *Figure 1—figure supplements 1–4*.

The following figure supplements are available for figure 1:

**Figure supplement 1**. Lack of indicated secondary structure in FG Nup constructs (panels **A**–**H**, text).

**Figure supplement 2**. Details of NMR relaxation data and derived correlation times.

**Figure supplement 3**. HSQC spectra of FSFG-K construct in buffer A vary with pH.

**Figure supplement 4**. Stability of FG Nup constructs by DLS.

*Figure 1. Continued*

**Figure supplement 5**. Stability of the FG-N construct in cell (left) and in buffer (right) by NMR.

**Figure supplement 6**. Transverse relaxation of FSFG-K in multiple environments.

proteins (IDPs), including FG Nups (*Uversky, 2009*; *Wang et al., 2011*; *Tetenbaum-Novatt et al., 2012*; *Phillip and Schreiber, 2013*), we mimicked the normal environment of NPCs using *Xenopus* egg extract, the best characterized environment for in vitro nuclear transport measurements (*Dabauvalle et al., 1991*). In addition, we tested the following: cytoplasm of living *Escherichia coli* using *in cell* NMR (*Serber et al., 2005*); *E. coli* high speed lysate (*Tetenbaum-Novatt et al., 2012*); and buffer alone, the latter lacking crowding agents or competitors and being the milieu in which these proteins have been most studied previously (*Frey et al., 2006*; *Lim et al., 2006*; *Ader et al., 2010*; *Yamada et al., 2010*).

We studied Nsp1, the most tested and characterized FG Nup which has been shown in vitro to mimic transport faithfully (*Jovanovic-Talisman et al., 2009*; *Hulsmann et al., 2012*). We focused on two segments of Nsp1 for which there is a consensus that they prototypically represent the extreme flavors and behaviors of FG Nups (*Yamada et al., 2010*); (i) the N-terminal segment of low charge (Asn-rich) and irregularly spaced FG repeats observed to be highly cohesive and form amyloid hydrogels under certain conditions (FG-N) (*Ader et al., 2010*) and (ii) the central segment of significant charge (Lys-rich) with regular FSFG repeats typically observed to be highly soluble (FSFG-K) (*Patel et al., 2007*; *Yamada et al., 2010*). The isolated N-terminal domain forms selective hydrogels in vitro (*Ader et al., 2010*) and is able to replace Nup98-FG domain in reconstituted *Xenopus* nuclei to provide the primary barrier enabling selective transport (*Hulsmann et al., 2012*). We studied these fragments individually and in combination in a wide range of constructs and conditions, including where the FG repeats are tethered to mimic their arrangement in the NPC (*Alber et al., 2007*) (*Table 1*, *Figure 1—figure supplement 1*).

## Results

### The cellular milieu maintains FG Nups as highly dynamic IDPs

FG Nups were fully disordered and highly dynamic in all cellular milieu tested. In NMR measurements, the degree of secondary structure is correlated with the difference between the $^{13}C$ chemical shift values of the α and β carbons relative to random coil values for each residue (*Schwarzinger et al., 2001*; *Tamiola et al., 2010*). The degree of secondary structure quantified in this way is near zero for all residues in FG-N and FSFG-K constructs in all conditions tested (*Figure 1B*, *Figure 1—figure supplement 1*) (*Eliezer, 2009*). This behavior of intrinsic disorder, judged from chemical shifts, is strikingly robust, being seen under a wide variety of conditions and also (for FSFG-K constructs) in buffers of varying pH (*Figure 1—figure supplement 3*).

Our observations in cellular milieux are in contrast to the behavior of the constructs containing the FG-N domain in buffer alone (*Hulsmann et al., 2012*). Under these conditions, dynamic light scattering and NMR measurements indicate that FG-N forms a hydrogel-like material (*Figure 1—figure supplement 4*). The in buffer behavior of our construct is consistent with extensive previous observations of β-sheet formation and aggregation of the N-terminal domain of Nsp1 (FG-N) (*Frey et al., 2006*; *Frey and Görlich, 2007*; *Ader et al., 2010*; *Labokha et al., 2013*). However, when FG-N constructs are observed by NMR inside a living cell, or in the presence of cell lysates or mimics, there is no appearance of high molecular weight components—NMR spectra of FG-N constructs in living *E. coli* did not change with time over more than 24 hr (*Figure 1—figure supplement 5*). These data indicate that the cellular milieu is a strong inhibitor of intermolecular FG repeat aggregation. The state of FG repeats in our system is therefore highly dynamic, as has been reported in vivo (*Mattheyses et al., 2010*).

To understand the differences between protein structure and dynamics in cellular mimics as compared to buffer, we measured spin relaxation parameters ($R_1$, $R_2$, nOe) which quantify the motion and interactions of the residues on timescales of μs to ms ($R_1$, $R_2$,) and ps to ns (nOe). NMR relaxation properties of FG Nups are significantly different between buffer alone and a protein-rich environment (*Figure 1C*). We measured large increases in $R_2/R_1$, indicative of transient spectral changes on interaction

**Table 1**. FG constructs prepared

| Abbreviation used in text | Simple structure | Sequence | Number of residues / monomer | MW | Nsp 1 ref start number | Data Figure |
|---|---|---|---|---|---|---|
| FSFG-K | M-Nsp1(274–397) | MDNKTTNTTPSFSFGAKSDENKAGA TSKPAFSFGAKPEEKKDDNSSKPAFS FGAKSNEDKQDGTAKPAFSFGAKPAE KNNNETSKPAFSFGAKSDEKKD GDASKPFSFGAKPDENKASATSKPA | 125 | 13,070 | 274 | 1, 1S1,1S2 |
| FSFG-K | M-Nsp1(274–397)-LEHHHHHH | MDNKTTNTTPSFSFGAKSDENKAGA TSKPAFSFGAKPEEKKDDNSSKPAFSF GAKSNEDKQDGTAKPAFSFGAKPAEKN NNETSKPAFSFGAKSDEKKDGDASKPAF SFGAKPDENKASATSKPALEHHHHHH | 133 | 14,135 | 274 | 1, 2, 1S1-6, 2S1-2 |
| FG-N | MGT-Nsp1(48–172)-SHMHHHHHH | MGTSAPNNTNNANSSITPAFGSN NTGNTAFGNSNPTSNVFGSNNS TTNTFGSNSAGTSLFGSSSAQQT KSNGTAGGNTFGSSSLFNNSTNS NTTKPAFGGLNFGGGNNTTPSST GNANTSNNLFGATASHMHHHHHH | 137 | 13,649 | 48 | 1, 1S1-2, 1S4-5, 2S1 |
| (FG-N)-(FSFG-K)-Tet-6His | MGT-Nsp1(48–172)-ASAT SKPA-Nsp1(284–397)-SHMGEYFTLQIRGRER FEMFRELNEALELKDA QAHMHHHHHH | MGTSAPNNTNNANSSITPAFGSNNTG NTAFGNSNPTSNVFGSNNSTTNTFGS NSAGTSLFGSSSAQQTKSNGTAGGNTF GSSSLFNNSTNSNTTKPAFGGLNFGGG NNTTPSSTGNANTSNNLFGATAASATS KPAFSFGAKSDENKAGATSKPAFSFGA KPEEKKDDNSSKPAFSFGAKSNEDKQ DGTAKPAFSFGAKPAEKNNNETSKPA FSFGAKSDEKKDGDASKPAFSFGAKPDE NKASATSKPASHMGEYFTLQIRGRERFE MFRELNEALELKDAQAHMHHHHHH | 292 | 30,235 | 48, 284 | 3, 4 |
| FSFG-K | MGTSATSKPA-Nsp1(284–397)-SHHHHHH | MGTSATSKPAFSFGAKSDENKAGATSKP AFSFGAKPEEKKDDNSSKPAFSFGAKS NEDKQDGTAKPAFSFGAKPAEKNNN ETSKPAFSFGAKSDEKKDGDASKPAFS FGAKPDENKASATSKPASHHHHHH | 131 | 13,721 | 284 | 4 |
| (FSFG-K)-(FSFG-K)-Tet-6His | MGTSATSKPA-Nsp1(284–397)-ATSKPA-Nsp1(284–397)-SHMGE YFTLQIRGRERFEMFRE LNEALELKDAQA HMHHHHHH | MGTSATSKPAFSFGAKSDENKAGATSKP AFSFGAKPEEKKDDNSSKPAFSFGAKSN EDKQDGTAKPAFSFGAKPAEKNNNETS KPAFSFGAKSDEKKDGDASKPAFSFGAK PDENKASATSKPASATSKPAFSFGAKSD ENKAGATSKPAFSFGAKPEEKKDDNSSK PAFSFGAKSNEDKQDGTAKPAFSFGA KPAEKNNNETSKPAFSFGAKSDEKKDG DASKPAFSFGAKPDENKASATSKPASHM GEYFTLQIRGRERFEMFRELNEALELKDA QAHMHHHHHH | 287 | 30,504 | 284 | 4 |
| (FG-N)-(FG-N)-Tet-6His | MGCT-Nsp1(48–172)-Nsp1(48–172)-SHMGE YFTLQIRGRERF EMFRELNEALELK DAQAHMHHHHHH | MGCTSAPNNTNNANSSITPAFGSNNTG NTAFGNSNPTSNVFGSNNSTTNTFGSN SAGTSLFGSSSAQQTKSNGTAGGNTFG SSSLFNNSTNSNTTKPAFGGLNFGGGN NTTPSSTGNANTSNNLFGATASAPNNT NNANSSITPAFGSNNTGNTAFGNSNPT SNVFGSNNSTTNTFGSNSAGTSLFGSSS AQQTKSNGTAGGNTFGSSSLFNNSTNS NTTKPAFGGLNFGGGNNTTPSSTGNA NTSNNLFGATASHMGEYFTLQIRGRER FEMFRELNEALELKDAQAHMHHHHHH | 296 | 29,927 | 48 | 4 |
| (FG-N)-(FSFG-K)-6His | MGT-Nsp1(48–172)-ASATSKPA-Nsp1(284–397)-SHHHHHH | MGTSAPNNTNNANSSITPAFGSNNTGN TAFGNSNPTSNVFGSNNSTTNTFGSNS AGTSLFGSSSAQQTKSNGTAGGNTFGS SSLFNNSTNSNTTKPAFGGLNFGGGNN TTPSSTGNANTSNNLFGATAASATSKPA FSFGAKSDENKAGATSKPAFSFGAKPE EKKDDNSSKPAFSFGAKSNEDKQDGT AKPAFSFGAKPAEKNNNETSKPAFSFG AKSDEKKDGDASKPAFSFGAKPDENKA SATSKPASHHHHHH | 257 | 25,955 | 48, 284 | 4 |

between the constructs and the cellular milieu (*Figure 1C*, *Figure 1—figure supplement 2*). The increases in $R_2/R_1$ were not seen upon increasing viscosity or crowding by inert agents (*Figure 1—figure supplement 6*), indicating that these changes result from weak binding of the FG repeats to the proteinaceous milieu as suggested by previous studies (*Tetenbaum-Novatt et al., 2012*). The phenylalanines and their adjacent residues show the primary interactions, with the greatest increases in $R_2/R_1$, while the spacer sequences remained relatively unaffected. The nuclear Overhauser effect (nOe) data show that the FG-N are highly mobile and flexible (*Figure 1C*, *Figure 1—figure supplement 2*), indicating no sequence-specific compaction, folding, or molten globule formation (*Uversky, 2009*; *Theillet et al., 2014*). The cellular milieu is thus in a state of constant, non-specific, dynamic interaction with the FG repeats.

Our results suggest that interactions of FG repeats with the cellular milieu stabilize the unfolded state by engaging the hydrophobic phenylalanines in transient interactions that decrease their contact with the water, reducing the driving force for hydrophobic collapse and amyloid formation. This is consistent with previous work that demonstrated that mutation of the Fs to more hydrophilic residues inhibits aggregation (*Frey et al., 2006*). Thus, exchange with the cellular milieu maintains the FG Nups in a highly dynamic, disordered state by inhibiting the intramolecular interactions that lead to aggregation in buffer.

## Random fuzzy interactions with transport factors

We used the atomic-level readout of our system to interrogate key aspects of FG Nup-TF interactions that have previously been inaccessible. It has long been established using crystallographic, computational, and NMR approaches that FG residues bind to hydrophobic pockets on TFs (reviewed in *Stewart, 2006*). However, the dynamics of this interaction and the behavior of the spacer regions between FG residues upon TF binding have remained uninterrogated, leading to proposals ranging from remaining disordered to significant structural rearrangements (*Rout et al., 2003*; *Peters, 2005*; *Frey et al., 2006*; *Lim and Deng, 2009*). Our assays provide a direct readout of the dynamics of the interaction and the state of the linker regions upon TF binding (*Figure 2*).

Many previous in vitro observations of FG-TF interactions lacked demonstration of reversibility and showed strong affinities (*Frey and Görlich, 2009*; *Tetenbaum-Novatt et al., 2012*; *Labokha et al., 2013*; *Kapinos et al., 2014*), incompatible with the rapid transport rates observed in vivo. By monitoring the recovery of the NMR spectral characteristics on dilution of Kap95, the interacting TF, we showed that the FG repeat-TF interactions are fully reversible in our system (*Figure 2—figure supplement 1*).

We used measured changes in amide resonances peak position and intensity as a function of Kap95 concentration using 2D Heteronuclear Single Quantum Coherence (HSQC) experiments (*Figure 2A*). Each (non-proline) residue contributes a peak, with the amide proton resonance frequency on the x-axis and nitrogen resonance frequency on the y-axis. The signal intensity of each peak is sensitive to the mobility and environment of that residue; upon binding to a TF, the effects of slower motion and of differing environment result in the signal of the corresponding peak decreasing. Strikingly, although the phenylalanine repeats themselves are attenuated upon Kap95 interaction, the majority of peaks are minimally affected, showing that the corresponding residues remain disordered and dynamic (*Figure 2A*). The spacer regions are thus highly mobile even within the interacting state. An extreme of this observation is seen for K332, which shows no attenuation (left subpanel in *Figure 2A*, lowest trace *Figure 2C*), and thus this residue remains flexible both in free and when bound to TF. When we examined attenuation as a function of residue number, we observed a striking periodic pattern of minimal attenuation for most residues and significant attenuation for FG repeats (*Figure 2B*). The degree of signal attenuation falls off relatively uniformly at both sides of each FG repeat, consistent with the transient binding of individual FG residues, with the remaining residues remaining fully disordered and highly dynamic. The minimally attenuated residues include those that have previously been implicated in forming phase changes of the FG repeat regions (*Ader et al., 2010*). We did not observe any major state change of spacers resulting from TF addition, such as caused by the formation or breakage of secondary structure, or gel/sol transitions (*Ader et al., 2010*; *Labokha et al., 2013*).

We observe that the binding interface between FG Nups and TFs at each interacting site is small—just the 2–4 residues surrounding the FG repeat itself—and the residues that do not interact directly with the TF sites remain highly mobile and dynamic. The size of the binding interface is consistent with other known transient interactions (*Clackson and Wells, 1995*; *Morrison et al., 2003*;

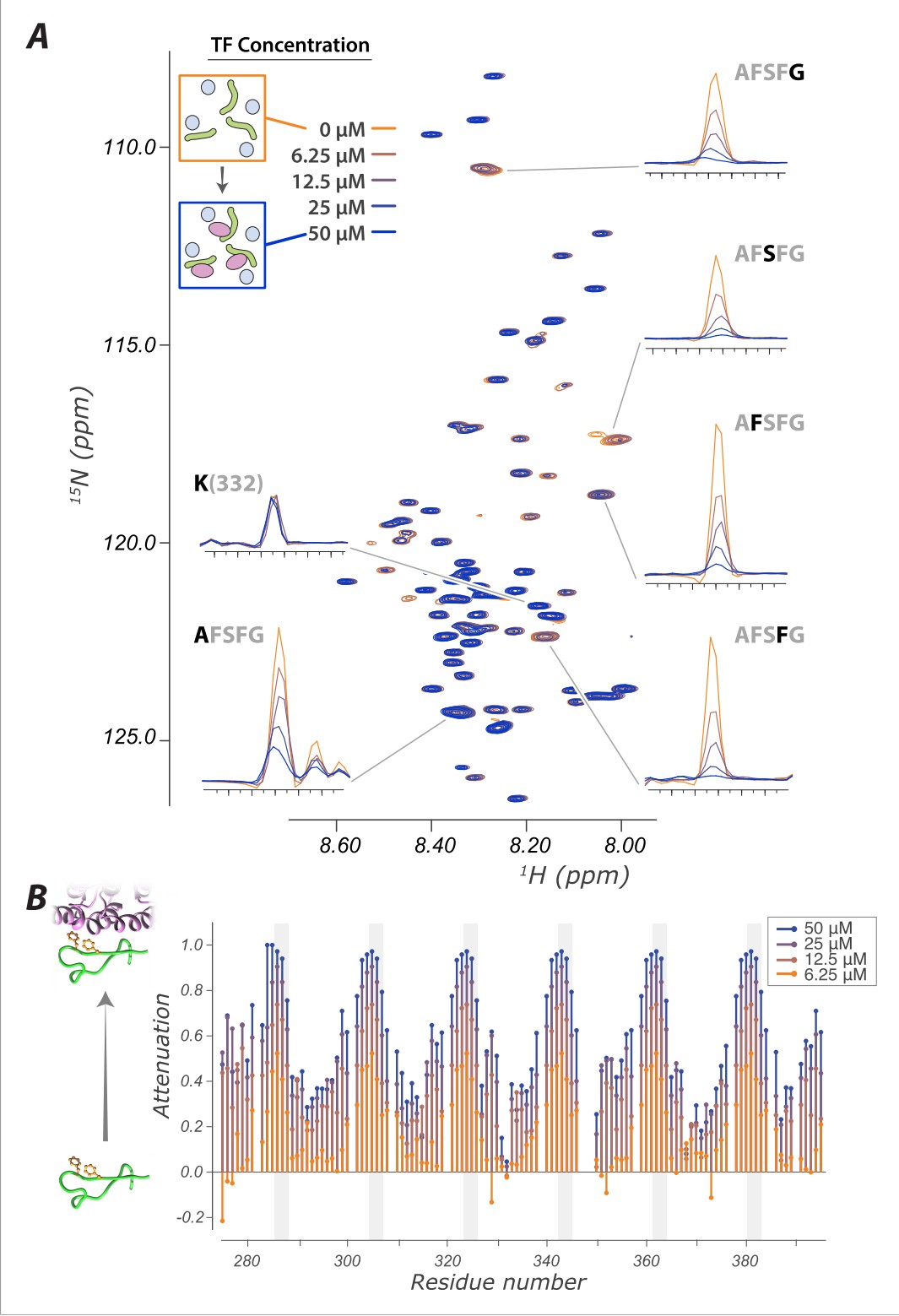

**Figure 2**. The interaction of FG Nups with the transport factor Kap95 is specific to the FG sites and leaves the spacers highly mobile (**A**) Overlay of spectra with varying concentrations of Kap95 (0, 6.125, 12.5, 25, 50 µM) in the presence of [U-$^{15}$N] FSFG-K (25 µM) in Xenopus egg extract, showing only the FG motifs strongly interact with the TF. Several peaks represent overlapping similar sequences with indistinguishable attenuation. (**B**) Superimposed

*Figure 2. continued on next page*

*Figure 2. Continued*

values of attenuation (($1 − I$)/$I_o$) on addition of Kap95 across the sequence; increased attenuation indicates a stronger interaction propensity for that residue (positions of FG motifs in the sequence are indicated by gray columns).

The following figure supplements are available for figure 2:

**Figure supplement 1**. Reversibility of TF/FG Nup interactions.

**Figure supplement 2**. $^{15}$N $R_2$ titration of Kap95 and [15N]FSFG-K.

*Tompa et al., 2009*; *Dixon et al., 2011*; *Ozbabacan et al., 2011*). We estimated a minimum effective affinity of Kap95 for the FSFG-K from a titration observing $^{15}$N $R_2$ (*Figure 2—figure supplement 2*) (*Su et al., 2007*). The true $K_d$ is greater than 36 μM, fully consistent with rapid, reversible transport.

Interaction or binding while remaining primarily disordered is highly unusual, as many functional IDPs are believed to adopt significant secondary structure formation upon interaction (*Wright and Dyson, 2009*; *Uversky and Dunker, 2013*), which we do not see here. This behavior is reminiscent of that proposed for random fuzzy complexes, in which one partner remains dynamically disordered and transient interactions predominate during selective recognition (*Sigalov, 2011*; *Fuxreiter and Tompa, 2012*).

## FG repeat behavior is largely independent of packing

Many observations suggest that there are multiple transport routes across the NPC, with FG Nups arranged in a variety of geometries (*Akey and Radermacher, 1993*; *Goldberg and Allen, 1996*; *Gant et al., 1998*; *Kiseleva et al., 2004*; *Alber et al., 2007*; *Burns and Wente, 2012*; *Laba et al., 2014*). Thus, because the NPC contains a large number of FG Nups in different arrangements to each other, we investigated the common features of the effects of local packing and attachment on FG repeats (*Figure 3*, *Table 1*). We designed chimeric proteins containing the FG domains in homotypic and heterotypic combinations. Some constructs included the tetramerization domain of p53, allowing us to form complexes that mimic the attachment of the FG domains to the NPC. We found essentially no change in chemical shifts and only modest changes in linewidth upon attachment to the p53 tetramerization domain (*Figure 3*). Any ordered protein of this size (~120 kDa) would be invisible using our NMR experiments, demonstrating that FG Nups remain dynamic IDPs even in large complexes, as previously measured for other large IDPs (*Kosol et al., 2013*). Notably, our largest constructs have a diameter measured by dynamic light scattering of 13 nm (*Figure 3B*), a significant fraction of the 30-nm diameter of the NPC. Our results reveal that the behavior of a given FG repeat is not affected strongly by the number or type of other FG repeats surrounding it (i.e., no 'emergent properties'). Our results also strongly indicate that there are no significant FG–FG interactions, as interactions between different types of FGs should impact their average environment. However, we cannot completely exclude that extremely weak, dynamic interactions can occur between FG repeats, and indeed such interactions may modulate the long distance distribution of FG repeat regions in the NPC. The FG Nups thus remain a highly dynamic fluid even at sizes, arrangements, and packing densities commensurate with the NPC. We observed the same pattern of rapid exchange of the FG residues with TF binding sites while the spacer regions remain highly mobile in all environments studied and with all constructs tested, showing the robustness of our findings (*Figure 4*).

While all overall behaviors are similar between FG-N and FSFG-K segments, in monomers or in tetramers, there are some intriguing differences. For example, in constructs where both FG types are present (*Figure 4*), not all FG-N phenylalanine residues are completely attenuated. This may be evidence for specificity of certain FG repeat types to binding particular TFs.

## Discussion

### FG Nups remain dynamic and disordered in their functional states

The FG repeats fill the central channel of the NPC to form a barrier to non-specific macromolecular diffusion (*Patel et al., 2007*; *Tetenbaum-Novatt and Rout, 2010*). We show that, even when densely

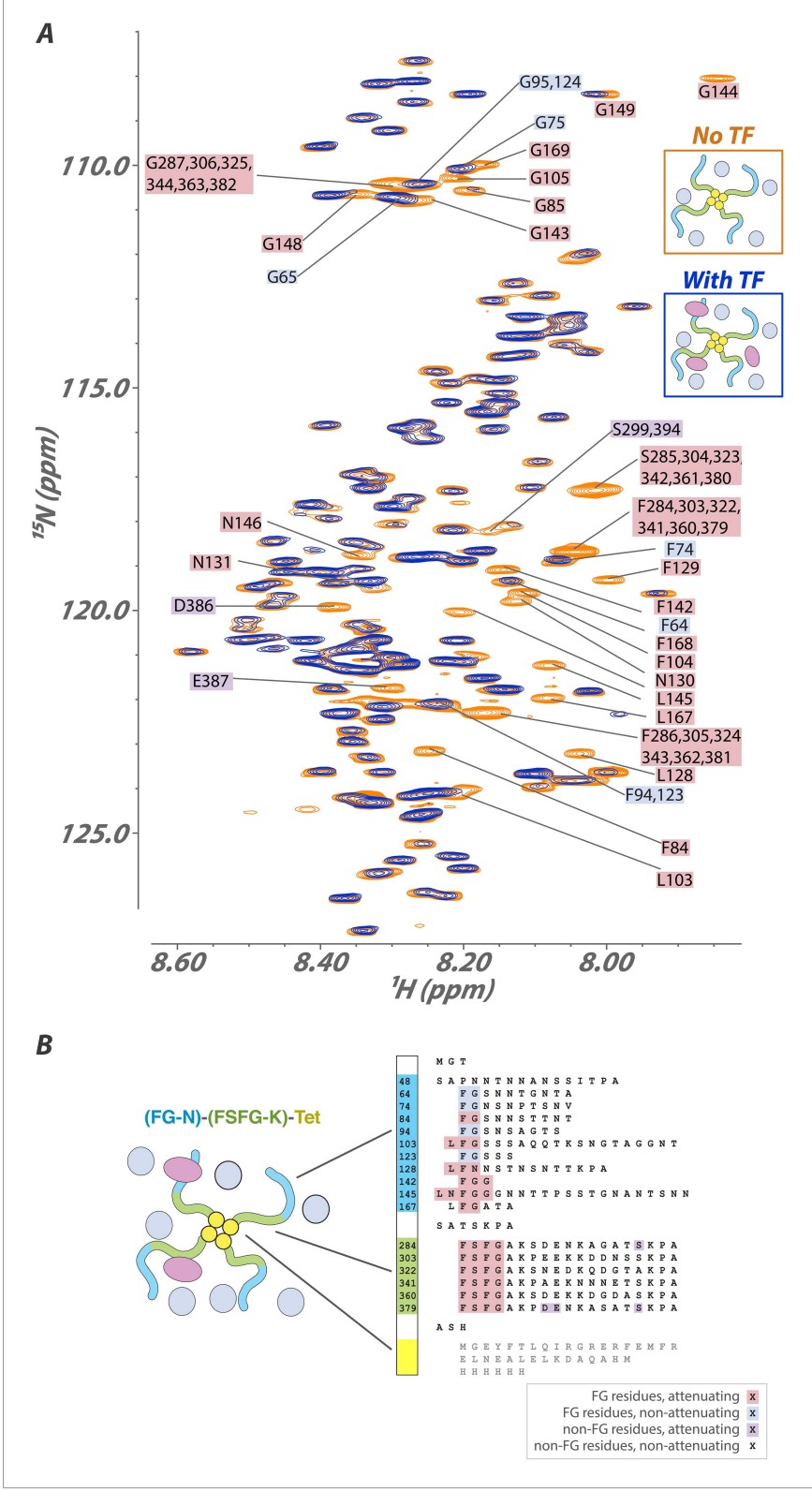

**Figure 3**. Interaction of a high molecular-weight tetramerized hybrid construct indicates that the binding mechanism is robust to changes in FG type, packing, and environment. (**A**) $^1$H[$^{15}$N] Heteronuclear Single Quantum Coherence (HSQC) spectra of the FG-N-FSFG-K-tet sequence (*Tables 1* and *2*) (MW 120 kDa) in the unbound (orange) and Kap95-bound (blue) states with *Xenopus* egg extract as the milieu. Peaks that undergo signal attenuation in the *Figure 3. continued on next page*

*Figure 3. Continued*

Kap95-bound spectrum indicate binding of those residues to Kap95. Typically, we observe that it is the FG residues (pink boxes), but not intervening spacer residues (no boxes), that attenuate upon binding to the transport factor Kap95; however, there are four FG repeats that are minimally attenuated (blue boxes), and a small number of spacer residues that are attenuated (purple boxes). (**B**) Schematic of the construct used, labeled as in (**A**). Very small chemical shift changes are observable for a few sites, consistent with rapid exchange into multiple inhomogeneous sites.

packed and in mixed flavors as in the NPC, the FG repeat regions studied here remain fully disordered and highly dynamic. The two domains studied represent the extreme behaviors observed for FG Nups in vitro (*Patel et al., 2007*; *Ader et al., 2010*; *Yamada et al., 2010*): FSFG-K is highly soluble and non-cohesive, while FG-N in buffer is highly cohesive, a prototypical hydrogel-forming FG Nup. It is clearly possible to make FG repeats aggregate (*Figure 1—figure supplements 4, 5*) (*Ader et al., 2010*), as has been demonstrated for many proteins (*Goldschmidt et al., 2010*). However, our data are inconsistent with any such hydrogel state in cellular milieu (*Frey et al., 2006*; *Labokha et al., 2013*). Re-arrangements of internal non-covalent crosslinks within hydrogels are on slow (second to minute) timescales, long enough to form a solid gel resistant to deformation, and too slow for solution NMR analysis (*Ader et al., 2010*). Instead of such a highly internally interacting, slowly moving gel, our results demonstrate that FG repeat regions form a highly dynamic phase, consistent with the rapid rates of nuclear transport. A highly dynamic, fluid state for FG repeats in vivo is in agreement with measurements of the living NPC (*Atkinson et al., 2013*). FG Nups do not appear to form a molten globule or collapsed state, as has also been proposed (*Yamada et al., 2010*), instead behaving as a fully disordered IDP (*Rout et al., 2003*; *Lim et al., 2006*; *Lim and Deng, 2009*; *Atkinson et al., 2013*). Our results also allow us to distinguish their IDP class (*Uversky, 2011*; *van der Lee et al., 2014*). FG repeats do not appear to populate more folded states either inter- or intra-molecularly upon interaction with either TFs or other FG Nups (*Wright and Dyson, 2009*; *Yamada et al., 2010*; *Han et al., 2012*; *Kato et al., 2012*). Instead, FG repeats lack any secondary structure, even when interacting with cognate binding partners, indicating that IDP mobility may be a key ingredient in nuclear transport.

## Non-TF interactions of FG repeats

Our results highlight the importance of the cellular environment for determining the behavior of the FG Nups because multiple weak interactions with cellular components stabilize the FG Nups in a disordered state. We propose that the selectivity of the NPC is thus maintained by both non-specific and specific cellular interactions, consistent with recent work demonstrating that FG permeability barrier is modulated by TFs (*Lim et al., 2007*; *Eisele et al., 2010*; *Milles et al., 2013*; *Kapinos et al., 2014*; *Lowe et al., 2015*; *Wagner et al., 2015*). We show that the modulation of permeability of FG brush behavior by TFs is not a result of structural changes of the FG Nups, but must result instead from the rapid, multivalent interactions of TFs with FG repeats (*Tetenbaum-Novatt et al., 2012*; *Schleicher et al., 2014*).

Our data, strongly supporting that FG repeats are highly dynamic tethered IDPs, imply that they must form entropic bristles (i.e., strongly sterically hindered regions) around their tether site (the central channel of the NPC) (*Rout et al., 2003*; *Strawn et al., 2004*; *Lim et al., 2006, 2007*; *Lim and Deng, 2009*; *Tetenbaum-Novatt and Rout, 2010*). This hindrance increases with increasing size of the passing macromolecules; hence, small proteins can pass more easily than large ones (*Tetenbaum-Novatt and Rout, 2010*). These larger nonspecific macromolecules interact with the FG repeats with insufficient frequency to overcome the entropic barrier of the FG repeats' polymer bristle structure, thus being effectively excluded (*Tetenbaum-Novatt and Rout, 2010*; *Ma et al., 2012*). TFs, however, have sufficient interaction frequency to do so—the original tenet of the 'virtual gating' idea, now delineated at the atomic scale by these results. In summary, the FG repeat regions, crowded around and within the central channel, may set up an entropic barrier that excludes macromolecules from their vicinity while permitting the approach of small molecules; however, macromolecules that interact with the FG repeats (such as TFs) can overcome this barrier (*Rout et al., 2003*; *Lim et al., 2007*) by their multiple and frequent interactions with FG Nups. We are pursuing experiments which we hope in the near future will gage the precise nature and magnitude of the exclusion mechanism.

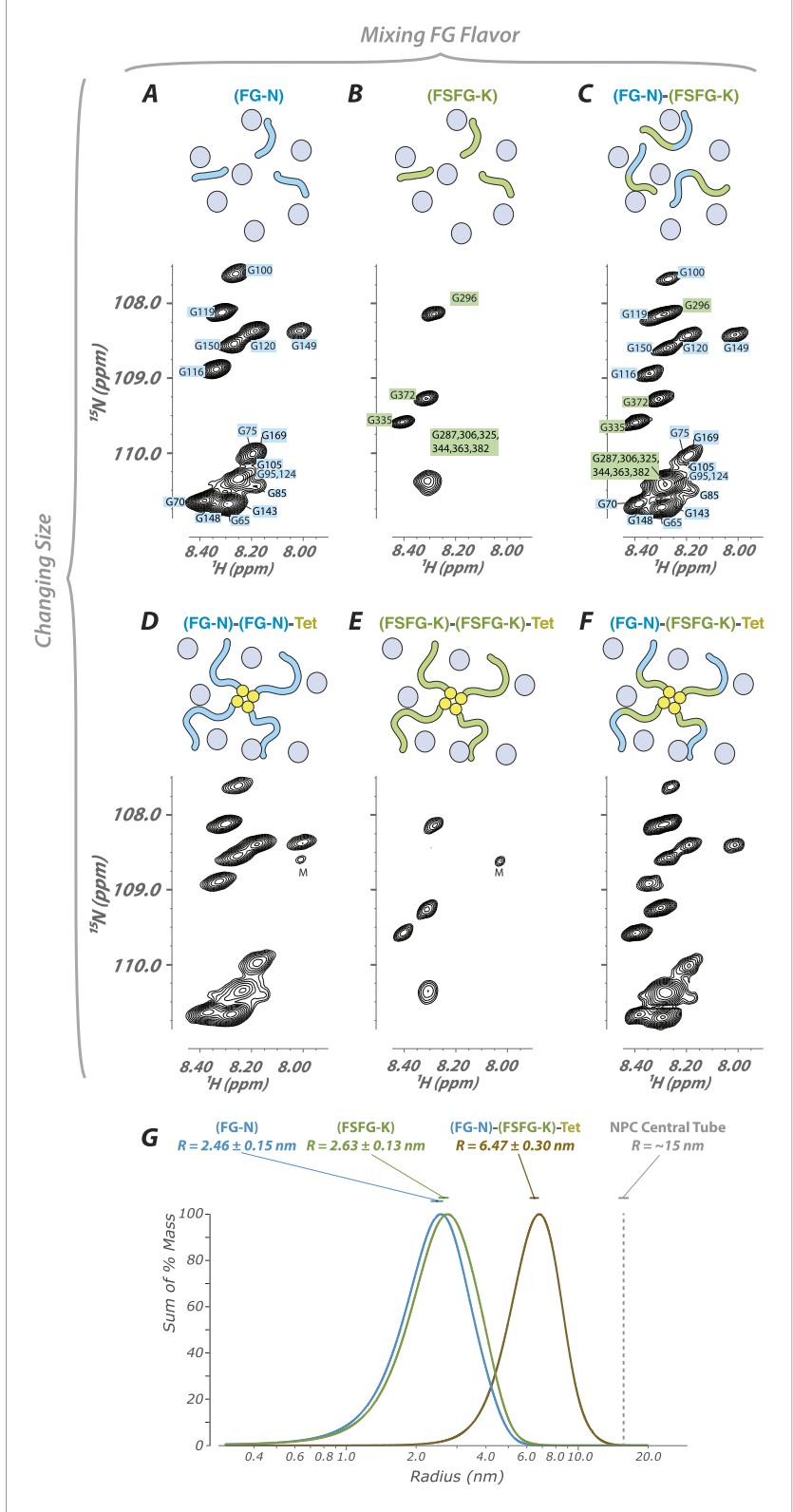

**Figure 4**. FG constructs show no significant FG–FG interactions, and remain fully disordered even when tethered at one end and packed at high density. (**A–F**) We developed a wide range of constructs containing fragments from Nsp1 (turquoise and green, numbering in *Figure 1*) and the small tetramerization domain from p53 (yellow). Schematics illustrate the different constructs as studied within the cellular milieu (blue circles) which vary both in total molecular weight, crowding of the FG repeats via tetramerization, and composition of FG fragments (homotypic or

*Figure 4. continued on next page*

*Figure 4. Continued*

heterotypic). Below each schematic is the HSQC spectrum of the glycine region of the represented construct observed in cell (*E. coli*). All constructs are tabulated in *Table 1*. 'M' identifies apparent $^{15}$N-labeled metabolites which are variable in intensity from preparation to preparation. (**G**) Dynamic light scattering of three representative constructs in buffer, showing homogeneity and consistency with expected size of constructs.

## Extremely fast on and off rates provide an explanation for rapid transport

Our results suggest that FG disorder is critical to the speed of selective nucleocytoplasmic transport. Selectivity is determined by the free energy gain upon TF-NPC binding, with the off rate (and so speed of transport) constrained by $k_{off} = K_d \cdot k_{on}$. For a given selectivity ($K_d$), the residence time within the NPC is constrained by the off rate. IDPs are able to engage far more rapidly than most ordered proteins (*Zhou, 2012*), allowing for high on and off rates while maintaining selectivity. For FGs in particular, several key lines of evidence indicate that the interaction of FG Nups with TFs is extraordinarily rapid, that is, each FG motif is in extremely fast exchange with TFs. We have directly shown that the binding interface is only 2–4 amino acids (the FG repeat itself). The buried surface area of this interaction is <1000 Å$^2$, consistent with other known transient interactions (*Clackson and Wells, 1995*; *Tompa et al., 2009*; *Dixon et al., 2011*). The spacer regions remain highly mobile and the degree of signal attenuation falls off relatively uniformly at both sides of each FG repeat (*Figure 2B*), fully consistent with the transient binding of individual FG residues. Within the NPC, a TF is able to rapidly diffuse, engaging transiently with multiple FG motifs across different FG Nups. The association rate of TFs with individual FG motifs is very fast because the FGs move very rapidly and the TF has multiple binding pockets available for interaction (*Bayliss et al., 2000, 2002*; *Bednenko et al., 2003*; *Isgro and Schulten, 2005, 2007*; *Liu and Stewart, 2005*). Taken together, these results indicate that the dynamics of FG-TF interaction are extraordinarily rapid. As a result, the interaction can be strong enough to be selective, and yet remain extremely fast, allowing rapid transit through the NPC.

In essence, it is the quantity, rather than quality, of interactions possible per molecule with FG repeats that distinguishes a TF from other macromolecules (*Figure 5*). Transient non-specific interactions of the

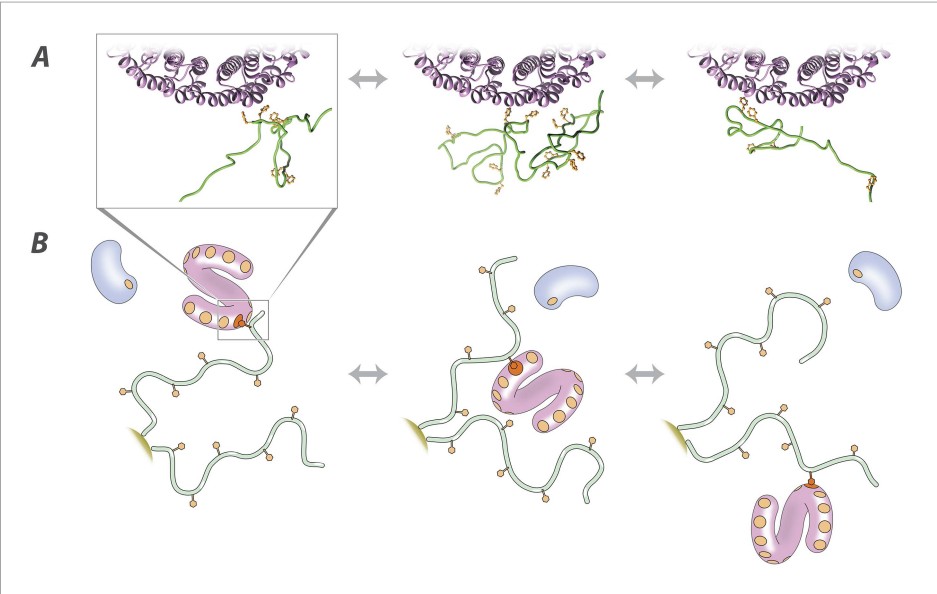

**Figure 5**. Molecular description of the nuclear transport mechanism. Specific conclusions from our results, as discussed in the main text, are illustrated with (**A**) details from a docked molecular simulation of the TF Kap95 (purple) and the FG repeat region FSFG-K, rendered in Chimera (see 'Materials and methods'), and (**B**) a diagrammatic view of the molecules involved. FG repeat (green; an ensemble of disordered conformers is illustrated in **A**). Phe (orange), Phe-binding pockets (**B**, orange circles), TF (pink), non-specific macromolecule (**B**, blue).

FG repeats with their milieu help maintain the disordered and dynamic nature of the barrier. Given the low sequence and structural complexity of the FG repeats, such non-specific interactions are to be expected but are not strong enough for these non-specific macromolecules to overcome the FG Nups' diffusion barrier (*Rout et al., 2003*). In contrast, TFs bind specifically to the FG motifs through pockets tuned for this purpose (*Stewart, 2003*, *2006*) while leaving the spacers between the motifs highly mobile. The fact that the spacers—and the FG repeats as a whole—remain fully mobile even when interacting with a TF also means that the barrier to non-specific passage formed by a dense fluid of FG repeats in the NPC remains fully intact, regardless of even large TF fluxes (a significant issue with models that invoke state changes, which our results and model circumvent).

While the FG repeats show little evidence of changed behavior upon oligomerization or TF interaction, it is likely that their structured attachment sites and diversity of flavors produce modulation of composition within the NPC. These variations are potentially important for the organization of alternate transport pathways (*Strawn et al., 2004*; *Terry and Wente, 2007*; *Yamada et al., 2010*). Our approach and its future extensions will thus be a powerful tool for the detailed characterization of these different nuclear transport pathways. The NPC is a biological example of the idiom 'many hands make light work'; the combined effect of many weak, transient interactions provides a specific multifunctional transport system, which allows for the simultaneous passage of hundreds of different macromolecules with a wide range of sizes and for robustness to significant alterations in the cell (*Wente and Rout, 2010*; *Adams and Wente, 2013*; *Tran et al., 2014*).

## Materials and methods

Proteins (*Table 1*) were expressed in BL21DE3 Gold cells using expression plasmids pET24a or pRSFDuet.

Protein concentration was determined by BCA analysis (Pierce Co.), except for Kap95 determined by $OD_{280}$ based on standard amino acid content. IPTG was from Gold Biotechnology. Bovine Serum Albumin (BSA), Pepstatin, PMSF, Protease Inhibitor Cocktail are from Sigma. *Xenopus* egg extract was the generous gift of Takashi Onikubo and David Shechter. Lyophilized *coli* extract was prepared by modification of the preciously described method (*Tetenbaum-Novatt et al., 2012*) using E. coli BL21 Gold cells grown in LB to $OD_{600}$ of 0.8, resuspended in Buffer A + 18 mg/l PMSF +0.4 mg/l pepstatin A, lysed by passing through a Microfluidizer, and centrifuged at 112,000 *g* for 30' at 4°C.

### Expression constructs and protein production

The FG-N, and FSFG-K segments were derived as previously described (*Tetenbaum-Novatt et al., 2012*) with the following modifications. Expression plasmids (pET24a, pRSFDuet) containing FG Nup fragments were transformed into BL21DE3 Gold cells (Agilent). Cells containing pET24a or pRSF constructs were grown to $OD_{600}$ of 0.8, induced with 1 mM IPTG and harvested after 2–4 hr. The cells' periplasm was removed by osmotic shock (*Magnusdottir et al., 2009*), and lysed under denaturing conditions (8 M urea). Urea was maintained throughout the purification for FG-N, whereas FSFG-K had 8 M urea in the lysis, 3 M urea in the first wash, and no urea in the remaining washes and elution. Proteins were purified on Talon resin in 20 mM HEPES, 150 mM KCl, 2 mM $MgCl_2$, (Buffer A, at pH 7.0), with protease inhibitor cocktail (PIC), PMSF, and pepstatin. Kap95 was prepared as previously described (*Tetenbaum-Novatt et al., 2012*) [U-$^{15}$N] and [U-$^{13}$C, $^{15}$N] materials were prepared using M9 media containing $^{15}NH_4Cl$ and [U-$^{13}$C] glucose (Cambridge Isotopes Limited, MA) as needed. The tetramerization domain of p53 was used to provide intramolecular crowding by construction of fusions with the tetramerization segment (*Poon et al., 2007*). As with all our constructs, the preparation was analyzed by size exclusion chromatography (SEC) to ensure that we were examining un-aggregated protein of the correct size and quantity (data not shown) and confirmed by dynamic light scattering.

### NMR measurements

Experiments were conducted on Bruker spectrometers at 700 MHz and 298 K unless otherwise indicated. In cell NMR methods used the general procedures developed for STINT-NMR (*Burz et al., 2006a*, *2006b*, *2012*), and other IDP studies. Assignment of resonances used the standard triple resonance approach as in *Sattler et al. (1999)*; *Muralidharan et al. (2006)*; *Tait et al. (2010)*; *Lemak et al. (2011)*; *Kalinina et al. (2012)*, including HNCO, HNCACO, HNCACM, CBCACINH, HNCA,

**Table 2**. Sequence assignments filed with BRMB (*Ulrich et al., 2008*)

| | |
|---|---|
| FGFG-K in cell *E. coli* expression | 25,182 |
| FSFG-K in buffer | 25,183 |
| FG-N in cell *E. coli* expression | 25,184 |
| FG-N in buffer | 25,185 |

HNCOCA, CCCONH, and HCCONH. Assignments were conducted in cell (*E. coli*)/buffer E and in buffer A, for the sets tabulated in *Table 2*, and were confirmed in other buffers by examination of HSQCs. Relaxation methods used standard procedures (*Barbato et al., 1992*; *Ferrage et al., 2006*, *2009*, *2010*). No corrections were applied for possible changes of fast exchange with solvent water, as described for $R_2$(CPMG) (*Kim et al., 2013*) for the following reasons. The observed changes for the IDP α-synuclein are less than two fold, and sequence-independent for a change of pH from 7.4 to 6.2 (*Figure 1*; *Kim et al., 2013*) while for FG Nups we observe up to c. sixfold sequence dependent variation of $R_2$. Secondly, the same sequence dependent variations of $R_2$ are seen in buffered *E. coli* extracts, with controlled pH (*Figure 1—figure supplement 6*). 'Titration' experiments were conducted at 800 or 700 MHz in buffers indicated, by preparation of separate samples for each change of concentration of Kap95.

Titration experiments for $^{15}$N $R_2$ were conducted at 500 MHz in buffer A, by preparing separate samples. Kap95 was fixed at 20 µM and $^{15}$N FSFG-K concentration was varied from 12.5 µM to 1200 µM. Each sample was prepared by addition of standard volumes of stocks to obtain the concentrations. No additional normalization was used. Each $R_2$ data acquisition used 6 delay points. For FSFG:Kap95 molar ratio of 1:1 and 0.625:1 the following delay times were used 0, 32.64, 65.28, 97.92, 130.56, 163.2 ms. For FSFG:Kap95 molar ratio of 4:1 the following delay times were used 0, 81.6, 163.2, 244.8, 326.4, 408 ms. For FSFG:Kap95 molar ratio of 7:1 the following delay times were used 0, 81.6, 163.2, 244.8, 326.4, 489.6 ms. For FSFG:Kap95 molar ratios of 60:1, 28.75:1 and 11:1 the following delay times were used 0, 81.6, 163.2, 326.4, 489.6, 652.8 ms. The same delays were used for an FSFG-N alone and the $R_2$ values obtained provided the value of $R_2$ free ($R_{2(0)}$) in the binding equation. The numbers of scans were optimized for S/N for each sample and total acquisition time per sample was ∼<13 hr.

## NMR analysis

The chemical shifts and their standard deviations (*Tamiola et al., 2010*) were used in the standard equation (*Schwarzinger et al., 2001*) to derive deviations from predicted shift values for the sequences of the assigned proteins. The values for these for $\Delta\delta^{13}C^{\alpha}$- $\Delta\delta$ $^{13}C^{\beta}$ are shown in *Figure 1—figure supplement 1*.

For NMR relaxation analysis, we derive approximate correlation times for ps/ns, associated with the different correlation times for hetero nuclear nOe by a direct calculation method that, in part, normalizes across different observation fields/frequencies, avoiding any complexities of spectral density analysis (for example, *Wirmer et al., 2006*), of the unverified applicability of a 'Model Free' approach (*Buevich et al., 2001*), and of analysis of simulated molecular dynamics which may not be generally applicable to all IDPs (*Kleckner and Foster, 2011*; *Xue and Skrynnikov, 2011*).

The apparent correlation times for motion of backbone amides was derived from use of the heteronuclear $^{15}$N[$^1$H] nOe using the standard dipolar treatment (*Neuhaus and Williamson, 2000*), ignoring chemical shift anisotropy contributions. For this pair, the field independent form from the standard formulae (p 135 in *Nicholas et al., 2010*) is given by

$$O_{noe} = 0.782804 - \frac{\left(0.010000683009 + 0.5132297763 \, (f_h\tau_n)^2\right)}{\left(0.002123332299 + 0.1102575629 \, (f_h\tau_n)^2 + \, (f_h\tau_n)^4\right)}, \quad (1)$$

where $f_H$ is the $^1$H frequency (Hertz) and $\tau_n$ is the apparent correlation time for the ns motion associated with direct dipolar relaxation (*Neuhaus and Williamson, 2000*; *Cavanagh et al., 2007*). We use here $\tau_n$ to discriminate from the overall correlation time for folded protein motion usually denoted as $\tau_c$.

$\tau_n$ may be directly obtained as

$$\tau_{n=} \sqrt{\left[\frac{(-(2.418830739.10^{30}O_{noe} + 2.02824096.10^{31}}{(4.056481921.10^{30}O_{noe} - 3.175430273.10^{31})}\right]/f_h}. \quad (2)$$

where the bracket numerator reads:
$$(0.005729037113O_{noe}^{2} + 0.07341631796O_{noe} + 0.2024235645)^{1/2} + 8.516066166.10^{30}$$

## Analysis of titration data

The data of *Figure 2* do not fit readily in most models of NMR titration for protein/protein interactions (*Goldflam et al., 2012*), especially for IDPs, where an intermediate folded form is frequent (for example, *Luna et al., 2014*; *Sugase et al., 2007*) In the case of slow exchange, it would be expected that at least part of the FG Nup sequence would show multiple peaks associated with some mobility in the 'bound' form. In the case of intermediate exchange, a very significant change in line shape associated with the exchange contribution to $^{1}$H $R_2$ would be expected. Since neither of these is observed, the spectra are interpreted as arising from fast exchange, with a minimal averaged chemical shift perturbation. This likely arises from multiple interaction sites and of ligand poses with a small average shift change, which does not exclude substantial shifts on interaction of individual sites or poses. The 'bound' form with comparable rotational tumbling to that of Kap95 would be expected to be broadened beyond detection for a $\tau_c$ of ~60 ns. The $^{1}$H line widths are only moderately increased suggesting modest $^{1}$H $R_{ex}$ contributions. On this basis, and subject to further investigation of exchange phenomena (for example, *Feeney et al., 1979*; *Kleckner and Foster, 2011*; Lepre et al., 2013; *van Dongen et al., 2002*) and of models of interaction (*Parks et al., 1983*; *Fielding et al., 2005*; *Simard et al., 2006*; *Wong et al., 2009*), we analyze the data as

$$R_{obs} = R_0(1-f) + R_b f, \quad (3)$$

where $R$ refers to observed, free (0) and bound (b) $^{15}$N $R_2$'s, and $f$ is the degree of occupancy on the TF for the specific residue (*Fushman et al., 1997*; *Su et al., 2007*). Note that the above equation assumes that any $R_{ex}$ additional contribution is negligible. This is justified if we assume that the actual $k_{ex}$ is very large, that is, fast (Equation 17b in *Lepre et al., 2004*). The back calculation of $K_d$ below then reflects a lower bound value subject to more precise, future determination of $R_{ex}$ terms. The relation of $R_{obs}$ to concentrations of FG Nup and TF then permits analysis of apparent equilibrium constants after *Cantor and Schimmel (1980)*, as

$$R_{obs} - R_0 = R_b\left(\left(K_d + L + T_f\right) - \left(\left(K_d + L + T_f\right)^2 - 4LT_f\right)\right)^{1/2}\bigg/2L, \quad (4)$$

where $T_f$ is the concentration of transport factor, and $L$ is the concentration of FSFG-K. From the observed data, *Figure 2—figure supplement 2*, the individual values of $K_d$ and $R_b$ are approximately determined. We used two methods which provided the same results within expected precision and experimental error. Using a MATLAB (Mathworks, Natick, MA) script, we produced a grid and found the minimum of conventional $\chi^2$ with the unknowns $R_b$ and $K_d$. The resulting grid was smooth with a single minimum (data not shown). In parallel, we used Prism Version 6.05 (GraphPad Software Inc, La Jolla, CA) with the above equation to obtain nonlinear regression curve fitting (*Motulsky and Brown, 2006*) to the data deriving $R_b$ and $K_d$. Since our assumptions of limited exchange contribution, site number equivalence of $L$ and $T_f$, and homogenous sites are significant simplifications, we then concentrate on the value of $K_d$ as a limiting lower value of the gross dissociation constant, estimated as 36± 6 µM. Per site of FSFG-K, for the six nominally equivalent FSFG motifs, this would be equivalent to an estimated lower value for a single site $K_d$ of 216 µM. Using Prism, we also derived sequence-specific apparent $K_d$ assuming that the FSFG residues sense 'different' equilibria. The values obtained–in µM (Std. Error) were Fsfg 31(6), fSfg 55(15), fsFg 44(9), and fsfG 17(6). The narrow range of values supports the assumption that $R_{ex}$ contributions are modest, since contributions are expected to be highly sequence dependent.

## Dynamic light scattering

Dynamic Light Scattering (DLS) measurements were made on a Dynapro Plate Reader (Wyatt Instruments, Santa Barbara, CA) at 298 K, in a 384 well plate with typically triplicate samples. Curves in *Figure 4G* are summations from instrument measurements. The mass-weighted averaged radii (and standard errors of the mean) are 2.46 (0.15) nm; 2.63 (0.14) nm; 6.47 (0.30) nm for FG-N, FSFG-K and FG-N-FSFG-K-tet, respectively. Concentrations used were 8, 4, and 2 mg/ml. Concentration dependencies were small. For the distribution curves the Dynapro software DYNAMICS 7.1.0.25 was used to analyze the experimental correlations with multiple 1 s runs, standard filtering including exclusions of <1% mass peaks, and regularization using the coil setting (*Hanlon et al., 2010*). The resulting binned regularized data was extracted and for each protein was summed across wells by scaled Gaussian summation at the peak position and width of the log-scale time base.

Note that cumulant analysis (*Koppel, 1972*) is not practical for multi-component systems with a large dynamic range. In each case the peak in the 1–10 nm regions represent >99% of the scattering mass. Averages and standard errors of the mean were calculated by pooling each mass-weighted average per well.

Figure 2—figure supplement 4 shows raw decay time data from DLS for FG–N and FSFG-K constructs in standard buffer A, 8 mg/ml. The vertical axis represents the % of mass having a specific decay time, with period in hours after start shown in the NE axis. The presentation of decay time avoids any issue of model for specific molecular shape or density or validity of the Raleigh-Debye approximation (*Berne and Pecora, 2000*). Data are pooled from a triplicate reading using the setting indicated above.

## Buffers used

A: 20 mM HEPES, 150 mM KCl, 2 mM $MgCl_2$, pH 6.5 unless otherwise noted;
B: *Xenopus* egg extract: protein egg extract prepared according to (*Shechter et al., 2004*) containing 31 mg/ml protein and was diluted with buffer A (3 + 10 v + v) with a final pH of 6.5;
C: *E. coli* lysate: lyophilized protein from high-speed spin dissolved in buffer A, at a final concentration of 30 mg/ml;
D: Buffer A plus 100 mg/ml BSA;
E: 20 mM Tris pH 7.5, 150 mM NaCl (Burz et al., 2006).

## Manipulated docking of IDP interaction with Kap95 (*Figure 4*)

HADDOCK, a rigid body docking algorithm (*Dominguez et al., 2003*), was used to combine Kap95 and FSFG-K into possible orientations to provide a molecular representation of our findings. Specifically, the crystal structure of Kap95 (3ND2) and a random conformations of FSFG-K was used as inputs and the FSFG-K repeat residues and residues on Kap95 identified by Isgro et al. (*Isgro and Schulten, 2005*) were chosen as active residues. Models were subsequently altered and images were rendered using Chimera (*Yang et al., 2012*).

## Acknowledgements

We are indebted to all members of the Rout and Cowburn Labs, and Dr Barak Raveh, Dr Geoffrey Armstrong, Professors Brian Chait, and Andrej Sali for discussion and support. *Xenopus* egg extract was the generous gift of Takashi Onikubo and Professor David Shechter.

## Additional information

### Funding

| Funder | Grant reference | Author |
| --- | --- | --- |
| National Institute of General Medical Sciences (NIGMS) | F32 GM087854 | Loren E Hough |
| Charles H. Revson Foundation | Fellow | Loren E Hough |

| Funder | Grant reference | Author |
|---|---|---|
| New York State Foundation for Science, Technology and Innovation (NYSTAR) | NYSBC grant | Kaushik Dutta, David Cowburn |
| U.S. Department of Defense (DOD) | 900 MHz NMR purchase | David Cowburn |
| National Institute of General Medical Sciences (NIGMS) | R01 GM06427 | Loren E Hough, Alia Kamal, Jaclyn Tetenbaum-Novatt, Michael P Rout |
| National Institute of General Medical Sciences (NIGMS) | U54 GM103511 | Loren E Hough, Alia Kamal, Jaclyn Tetenbaum-Novatt, Michael P Rout |
| National Institute of General Medical Sciences (NIGMS) | R01 GM071329 | Loren E Hough, Alia Kamal, Jaclyn Tetenbaum-Novatt, Michael P Rout |
| National Institute of General Medical Sciences (NIGMS) | P41 GM066354 | David Cowburn |
| National Institutes of Health (NIH) | C06 RR015495 | David Cowburn |
| W.M. Keck Foundation | 900 MHz NMR purchase | David Cowburn |
| National Institutes of Health (NIH) | 1S10OD016305 | David Cowburn |

The funders had no role in study design, data collection and interpretation, or the decision to submit the work for publication.

### Author contributions
LEH, Conception and design, Acquisition of data, Analysis and interpretation of data, Drafting or revising the article; KD, SS, DBT, Acquisition of data, Analysis and interpretation of data, Drafting or revising the article; AK, JT-N, Acquisition of data, Drafting or revising the article; MPR, DC, Conception and design, Analysis and interpretation of data, Drafting or revising the article

### Author ORCIDs
Loren E Hough, http://orcid.org/0000-0002-1104-0126
Kaushik Dutta, http://orcid.org/0000-0002-9653-3842
David Cowburn, http://orcid.org/0000-0001-6770-7172

# Additional files

### Major datasets
The following datasets were generated:

| Author(s) | Year | Dataset title | Dataset ID and/or URL | Database, license, and accessibility information |
|---|---|---|---|---|
| Hough LE, Dutta K, Sparks S, Temel DB, Kamal A, Tetenbaum-Novatt J, Rout MP, Cowburn D | 2014 | FGFG-K in cell E. coli expression NMR assignments | http://www.bmrb.wisc.edu/data_library/summary/index.php?bmrbld=25182 | Publicly available at the BMRB Biological Magnetic Resonance Data Bank (Accession no: 25182). |
| Hough LE, Dutta K, Sparks S, Temel DB, Kamal A, Tetenbaum-Novatt J, Rout MP, Cowburn D | 2014 | FSFG-K in buffer NMR assignment | http://www.bmrb.wisc.edu/data_library/summary/index.php?brmbld=25183 | Publicly available at the BMRB Biological Magnetic Resonance Data Bank (Accession no: 25183). |
| Hough LE, Dutta K, Sparks S, Temel DB, Kamal A, Tetenbaum-Novatt J, Rout MP, Cowburn D | 2014 | FG-N in cell E. coli NMR assignment | http://www.bmrb.wisc.edu/data_library/summary/index.php?bmrbld=25184 | Publicly available at the BMRB Biological Magnetic Resonance Data Bank (Accession no: 25184). |

| Author(s) | Year | Dataset title | Dataset ID and/or URL | Database, license, and accessibility information |
|---|---|---|---|---|
| Hough LE, Dutta K, Sparks S, Temel DB, Kamal A, Tetenbaum-Novatt J, Rout MP, Cowburn D | 2014 | FG-N in buffer NMR assignment | http://www.bmrb.wisc.edu/data_library/summary/index.php?bmrbld=25185 | Publicly available at the BMRB Biological Magnetic Resonance Data Bank (Accession no: 25185). |

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
