## [Decision Letter]

Thank you for submitting your work entitled “The molecular mechanism of nuclear transport revealed by atomic scale measurements” for peer review at *eLife*. Your submission has been favorably evaluated by John Kuriyan (Senior Editor) and three reviewers, one of whom, Volker Dötsch, is a member of our Board of Reviewing Editors. The other two reviewers, Masahiro Shirakawa and Miquel Pons, have agreed to share their identity.

The reviewers have discussed the reviews with one another and the Reviewing Editor has drafted this decision to help you prepare a revised submission.

The paper describes the physicochemical properties and functions of a major region of Nsp1 protein, one of the major components of Nuclear Pore Complex (NPC). The region mainly consists of phenylalanine-glycyl (FG) repeats (FG Nups) and spacer amino acid sequences between them. It is thought to be an intrinsically denatured protein (IDP) region, and bundles of them function as a filter for transporter factors and barriers for non-specific diffusion, as a polymer brush. The authors show that FG Nups maintain IDP properties even at high concentration or when forming multimeric architectures or under the cellular milieu, by smart use of solution NMR. They also showed that the FG repeats of FG Nups selectively interact with transporter factors, but these interactions seem to cause no structural change on FG Nups, because both *k*_*on*_ and *k*_*off*_ are small, and thus each contact time is too short to induce conformational changes. Thus, although in general IDPs are believed to adopt secondary structure upon interaction, FG Nups interact as if they are in random and dynamic conformation. This property is used to explain how transporter factors pass through the filter rapidly. This seems to be the most interesting finding reported in this manuscript.

Essential revisions:

1) The components of the “cellular milieu” preventing aggregation seem to be present in *Xenopus* egg extracts, inside *E. coli* cells and in *E.coli* high speed spin. The last condition is described as based on “lyophilized protein”. Can it be defined more specifically? Why such a high protein concentration is needed (230 mg/ml Figure 1—figure supplement 6)? Presumably that exceeds that of the intact *E. coli* cytoplasm. Could it be that other non-protein components (e.g. lipids) also play a role? The FGGF sequence forms the core of the Unique Lipid Binding Region (ULBR) in the intrinsically disordered domain of c-Src.

2) Normally, the influence of proteins in the cellular environment is investigated by repeating the experiments in the presence of BSA or other inert proteins. The authors report control experiments for general viscosity effects but such a protein specific experiment is missing.

3) In the pH dependent spectra, in addition to the attenuation due to solvent exchange, there is at least one residue that is titrating. Could it be from the His tag? Please confirm.

4) The periodic change in *R*_*2*_ around FG repeats in vitro and the equivalent change in signal attenuation in the presence of Kap95 is interpreted assuming that exchange contributions are negligible. In the absence of “rescuing” interactions, clustering of FX regions is certainly occurring (it is much easier than intermolecular interactions and those are readily observed as hydrogel formation). Therefore it is not impossible that the observed signal attenuation or *R*_*2*_ effects are caused by a decrease in the exchange rate between intramolecular FG–FG interactions. Otherwise, why is the *R*_*2*_ periodicity not observed in NOEs? The discussion on analysis of the titration data could probably be expanded. It raises some questions about the derived *K*_*d*_ values.

5) Some clarification on Figure 2—figure supplement 2 is necessary. It seems that each point (e.g. FSFG) is the average *R*_*2*_ of all distinguishable sites and that the error bars are the standard deviation of the *R*_*2*_ values of individual sites. This should be clarified. Normally standard deviations refer to multiple measurements on the same site.

If individual sites are measured and the interaction is non-cooperative, then the measured *K*_*d*_ would be the one for each site and it is unclear how the estimate of 216 μM per site is derived. Please clarify.

6) Please define “attenuation” used in Figure 2 (e.g 1–Ia/Ib). Because the spectra are from independent samples, there is probably some normalization.

7) What is the mechanism for larger proteins to be excluded from passing and smaller proteins being allowed to pass the nuclear pore? Is it just steric hindrance that is overcome by specific and weak interactions of the transport factors? While the paper focuses mainly on the interaction with the TFs, more discussion of the model of excluding all other proteins would be helpful.

---

## [Author Response]

*1) The components of the “cellular milieu” preventing aggregation seem to be present in Xenopus egg extracts, inside* E. coli *cells and in* E.coli *high speed spin. The last condition is described as based on “lyophilized protein”. Can it be defined more specifically?*

The preparation of the lyophilized powder is now expanded in the text.

*Why such a high protein concentration is needed (230 mg/ml*
Figure 1—figure supplement 6*)? Presumably that exceeds that of the intact* E. coli *cytoplasm.*

The high concentration of protein extract used was based on several published estimates of ∼30% w/v (e.g. Cayley 1991) *in E. coli*. We added a note to the legend of Figure 1—figure supplement 6.

Could it be that other non-protein components (e.g. lipids) also play a role? The FGGF sequence forms the core of the Unique Lipid Binding Region (ULBR) in the intrinsically disordered domain of c-Src.

We agree that it could be that other components could play a role, which is why we largely refer to it as the cellular milieu. However, the robustness of our findings to a variety of conditions (i.e. high speed lysates, cellular milieu, *Xenopus* extracts) and purified protein (BSA, see below), each of which differ in their protein/DNA/RNA/lipid compositions, suggest that it is the protein playing the dominant role. In addition, in previous work, the effects of removing protein or other components indicated that proteins were most important for modulating the FG-TF binding affinity. This study largely excluded that the effect arises from either small molecules (including lipids) or nucleic acids: “the effect was likely due to protein rather than small molecules or nucleic acids (supplemental Fig. 10)” (116). In this work, classical heating procedures were used to remove proteins – but not nucleic acids, small molecules, and lipids – from a cell lysate. Likewise, the lysate was passed through a low molecular weight (10 kD) cutoff filter. In both cases, the competition effect was removed. Taken together these data strongly indicated that it was macromolecules and most likely proteins, rather than nucleic acids or small molecules such as lipids, that accounted for the great majority of the competition effect. An additional reference to this is added in the subsection “The cellular milieu maintains FG Nups as highly dynamic IDPs ”.

2) Normally, the influence of proteins in the cellular environemnt is investigated by repeating the experiments in the presence of BSA or other inert proteins. The authors report control experiments for general viscosity effects but such a protein specific experiment is missing.

We agree, and so performed these experiments in the presence of BSA. We found that BSA had a significant effect on the relaxation properties of FSFG, rather than BSA being an inert protein. This result is expected, consistent with our observations above that proteins in lysate contribute the majority effect, and the known interactions of BSA with aromatic amino acids. The modified Figure is Figure 1—figure supplement 6.

3) In the pH dependent spectra, in addition to the attenuation due to solvent exchange, there is at least one residue that is titrating. Could it be from the His tag? Please confirm.

Yes, that is the His tag. We have clarified the legend to Figure 1—figure supplement 3 and added to the legend to describe the titration. The specific resonance is now identified in the figure.

*4) The periodic change in R2 around FG repeats* in vitro *and the equivalent change in signal attenuation in the presence of Kap95 is interpreted assuming that exchange contributions are negligible.*

We do not explicitly ascribe a common single mechanism to the in vitro and the Kap95 R2 changes.

In the absence of “rescuing” interactions, clustering of FX regions is certainly occurring (it is much easier than intermolecular interactions and those are readily observed as hydrogel formation). Therefore it is not impossible that the observed signal attenuation or R2 effects are caused by a decrease in the exchange rate between intramolecular FG–FG interactions.

The concatenation and tetramerization of FG Nups shown in Figure 4 strongly indicates that there are no significant FG–FG interactions, because such interactions between different types of FGs should impact their average environment. Our results can rule out gross state changes such as hydrogel formation; however, we can directly neither rule out nor support the suggestion that the FX repeats are very weakly dynamically clustered (weak cohesive interactions) or that this very weak clustering is altered upon FX-lysate interaction. Importantly, though, this does not affect the main conclusions of our paper that the FX repeats themselves are highly dynamic. We add a comment in the subsection “FG repeat behavior is largely independent of packing”.

We believe the interpretation that the FG-lysate effects are primarily due to FG-lysate interactions is the most parsimonious explanation given all of our experiments, and is supported by several results. First, significant changes in FG–FG interactions could induce significant changes in spacer conformation, which we do not see. Second, in previous work (116), lysate reduces FG Nup-TF affinities, indicating that lysate (through nonspecific interactions) are occupying FG repeats (116). Third, if interactions with the cellular milieu resulted in an increase in the FG–FG exchange rate, we would also expect to see a similar effect on increasing of viscosity or crowding by PVP, which we did not (Figure 1—figure supplement 6).

Otherwise, why the R2 periodicity is not observed in NOEs?

As for the difference in effects seen in cell for the R2 vs NOE, we agree that chemical exchange phenomenon must be at play arising from nonspecific interactions in cell. We observe that the ps-ns effects on the heteronuclear nOe are not substantially changed on non-specific interactions. A hypothesis then arising is that the interactions are very dynamic and contact times of single specific interacting conformers are very short. Since there is little precedent for such a situation, we suggest that this be explored further elsewhere, and we limit our speculation on the matter.

*The discussion on analysis of the titration data could probably be expanded. It raises some questions about the derived* K_d_
*values.*

We have added language to the Appendix, Section 2.3 to clarify our assumptions in deriving a *K*_*d*_ value (see the legend for Figure 2—figure supplement 2).

*5) Some clarification on*
Figure 2—figure supplement 2
*is necessary. It seems that each point (e.g. FSFG) is the average R*_*2*_
*of all distinguishable sites and that the error bars are the standard deviation of the R*_*2*_
*values of individual sites. This should be clarified. Normally standard deviations refer to multiple measurements on the same site.*

*If individual sites are measured and the interaction is non-cooperative, then the measured* K_d_
*would be the one for each site and it is unclear how the estimate of 216 μM per site is derived. Please clarify.*

We have added substantial detail of how the error estimates were made in Appendix Section 2.3. The standard errors are calculated by the Prism program as part of the nonlinear fit statistical analysis, and reflect the equivalence of the multiple observations of the variables, affinity, and bound *R*_*2*_ value from the different sites, rather than from repeated measurements.

*6) Please define “attenuation” used in*
Figure 2
*(e.g 1–Ia/Ib).*

Added to Figure 2 legend.

Because the spectra are from independent samples, there is probably some normalization.

The independent samples were all made at the same concentration of FG, with varying TF concentration, by addition of stock Kap95 and/or buffer to a constant volume of FG Nup. Therefore, no normalization was needed.

7) What is the mechanism for larger proteins to be excluded from passing and smaller proteins being allowed to pass the nuclear pore? Is it just steric hindrance that is overcome by specific and weak interactions of the transport factors? While the paper focuses mainly on the interaction with the TFs, more discussion of the model of excluding all other proteins would be helpful.

Yes, we agree – it appears to be the steric hindrance of the FG repeat IDPs, which set up “entropic bristles” (Hoh, 1998; [91]; [96]; Tompa, 2005; [108]; [115]). We did not enter a detailed discussion regarding the exclusion mechanism, as the experiments described in this manuscript were designed mainly to address the conundrum of how both selective and rapid exchange of specific transport factors can be achieved. As requested, we have expanded our discussion of these points in the subsection “Non-TF interactions of FG-repeats”.

References: Cayley S, Lewis B, Guttman H, Record Jr M. Characterization of the cytoplasm of Escherichia coli K-12 as a function of external osmolarity. Implications for protein-DNA interactions in vivo. Journal of molecular biology. 1991;222(2):281.

Hoh, J. H. Functional protein domains from the thermally driven motion of polypeptide chains: a proposal. Proteins 32, 223–228, doi:10.1002/1097-0134 (1998).

Tompa, P. The interplay between structure and function in intrinsically unstructured proteins. FEBS Lett 579, 3346–3354, doi:10.1016/j.febslet.2005.03.072 (2005).